# An Amplitude-Modulated Metadevice with Switchable Reflection, Transmission, and Absorption

**DOI:** 10.3390/nano12234227

**Published:** 2022-11-28

**Authors:** Sheng Ye, Chengye Huang, Jinglin He, Hanru Shao, Minhua Li, Jianfeng Dong

**Affiliations:** Faculty of Electrical Engineering and Computer Science, Ningbo University, Ningbo 315211, China

**Keywords:** reconfigurable metadevice, microstructure, PIN diodes, coding, equivalent circuit model

## Abstract

In this paper, we propose a reconfigurable metadevice with independent polarization control based on a 90° rotationally symmetric microstructure. Three functionalities of broadband high-efficiency transmission, broadband high-efficiency reflection, and perfect absorption are switched by the on-state and off-state PIN diodes. Coding metadevices designed with diversified lumped element combinations are further studied in detail. By controlling the two diodes on the top layer in opposite states, absorption bandwidth is significantly improved. Reasonable arrangements of coding sequences allow for reflected dual/multi-beam modulation. Electric field distribution, power loss, complex impedance functions, and equivalent circuit models are used to better analyze the physical mechanism of the design. A prototype of the microstructure has been fabricated, and the experimental results agree well with the simulation. Electronic components integrated microstructures with high degrees of freedom have potential applications in intelligent wireless communication, electronic detection, advanced sensors, and smart stealth radomes.

## 1. Introduction

Metamaterials (MMs) [1,2] are periodic or aperiodic artificial materials composed of subwavelength metal/dielectric microstructures. Due to their unusual physical characteristics, many exciting phenomena have emerged, such as the negative refractive index [1,2], light polarization control [3], invisibility cloak [4], holographic imaging [5], circular dichroism [6,7], etc. Metadevices (Metasurfaces) [8,9] are the extension and development of MMs, consisting of single or ultrathin multilayer microstructures arranged in two dimensions, with the advantages of easy processing and integration.

Generally speaking, passive metadevices have a fixed topological geometry. In other words, once the structure-only metadevices have been fabricated, their functionalities will be fixed and cannot be changed in real-time to meet the requirements. In some practical applications, the ability to switch between different functionalities is often demanded. Therefore, active metadevices have attracted considerable attention from scholars, which consist of two main concepts: “tunable” and “reconfigurable”. Materials with dynamic optical performance are integrated into the metadevice for versatility, such as Graphene [10,11], GST [12], VO_2_ [13,14], etc. Its optical and electrical properties will change dramatically during the phase transition. Hence, it can be used as thermally or electronically switchable components. Additionally, some active devices can also be integrated into the metadevices in the microwave regime, such as PIN diodes [15,16], varactors [17,18], etc. By controlling the external excitation signal to adjust the radiofrequency (RF) characteristics of PIN diodes, a fixed form of metadevice can exhibit dynamically tunable or reconfigurable multifunctional electromagnetic (EM) properties.

When incident waves encounter the material surface during their propagation process, they will be divided into three parts: reflected waves, absorbed waves, and transmitted waves. Metadevices can be used to modulate the amplitude of EM waves with total reflection, high-efficiency transmission, and perfect absorption being the three main functionalities. Zhu et al. [19] proposed a switchable reflector/absorber which can be continuously controlled from total reflection to perfect absorption. It can also react selectively to different polarized waves. Based on active frequency selective surface (AFSS), Ghosh et al. [20] proposed a switchable reflector/absorber. It has narrow/wide-band absorption characteristics. Li et al. [21] implemented a real-time switchable reflector/absorber, and the proposed structure covers a much wider frequency band compared to previous designs. Lustrac et al. [22] proposed an EM screen with switchable transmission, reflection, and absorption functionalities. It switches the three modes by changing the states of diodes with on, off, and resistive, but the harsh current triggering conditions and size limit its practical application. Works on the possibility of switching among transmission, reflection, and absorption utilizing a single metadevice at the same frequency are less, with most research being carried out for only two functionalities: absorption/reflection [19,20,21,23,24], absorption/transmission [25,26,27,28], or transmission/reflection [29,30,31]. Additionally, the limitations of operating bandwidth and efficiency prevent it from being widely used. For these reasons, the wider frequency band and high efficiency of these three functionalities on the metadevices are greatly considerable.

To solve the above problems, an amplitude-modulated metadevice with switchable broadband high-efficiency transmission, broadband high-efficiency reflection, and perfect absorption is presented. The microstructure consists of two controllable sections loaded with PIN diodes, and EM responses in the orthogonal direction do not interfere with each other. The physical mechanisms of the three modes are revealed by analyzing the electric field, power loss, different loss tangents, and complex impedance functions. Furthermore, the equivalent circuit is used to optimize the parameter design of the structure. At last, we investigated the effects of oblique incidence and various combinations of lumped elements.

## 2. Materials and Methods

The conceptual diagram of the amplitude-modulated metadevice is illustrated in Figure 1a. The meta-atom consists of four metallic films and two substrates. Figure 1b shows the perspective view of a meta-atom which can be divided into two parts: part A and part B. Both part A and part B consist of a double-sided metallic structure. Taking part A as an example, it is composed of the first layer of I-shaped metallic film, the second layer of dielectric substrate Rogers RO4003C (a dielectric constant of 3.38 + i0.0027), and the third layer of I-shaped metallic film (a 90° rotational symmetry with the first layer). An air spacer layer with a thickness of 4.5 mm is introduced between part A and part B. By adjusting the thickness of the air spacer, three switchable functionalities can be realized in the same frequency range. It is worth mentioning that due to the fourfold rotational (C4) symmetry of the structure, the metadevice can simultaneously generate two independent operating functionalities covering its two sides under the illumination of 45° polarized incident waves. In other words, the metadevice has independent EM responses to x-polarized and y-polarized waves by independently controlling vertically-loaded diodes (PIN diode-1 and PIN diode-3) and horizontally-loaded diodes (PIN diode-2 and PIN diode-4). Changing the states of PIN diodes in the cross-polarization direction has no effect on the resonance response, which indicates that the metadevice has a high cross-polarization isolation effect [17]. In the design of the bifunctional metadevices with reflection and absorption, there is no transmission due to the consecutive metal shielding at the bottom for simplifying the design. To add the functionality of transmission, referring to the frequency selective surface [30], we replaced the metal ground with a microstructure loaded with PIN diodes. On this basis, this complex task is completely simplified by designing two different meta-atoms that allow for absorption controlled (part A) and transmission-reflection controlled (part B), which are then combined to form the desired meta-atom. To achieve a tri-functional switching/tunable effect, it is necessary to apply a bias voltage to regulate the lumped parameters of PIN diodes. The direct current (DC) circuit of PIN diodes and feed source can be connected via radiofrequency (RF) chokes, which obstruct the RF signals to ensure good isolation between the DC and RF performances [16]. At the same time, the bias network is carefully optimized, which has almost no effect on the initial performance. Here, the metallic layers are made of copper (a conductivity of σ = 5.8 × 10^7^ S/m) with a thickness of 0.035 mm. Figure 1d,e show the geometric parameters after optimization: P_x_ = 10.1 mm, P_y_ = 10.1 mm, h1 = 0.813 mm, h2 = 0.813 mm, g1 = 4 mm, g2 = 1.1 mm, g3 = 1.1 mm, g4 = 2.6 mm, w1 = 7 mm, w2 = 2.1 mm, w3 = 7 mm, w4 = 0.2 mm, and w5 = 2.5 mm. The type of PIN diode here is the BAP50-03 [32] of NXP semiconductors. As shown in Figure 1c, the PIN diode is modeled as series R_off_-C_off_ (without biased voltage, R_off_ = 40 Ω, and C_off_ = 0.19 pF) and R_on_-L_on_ (with forward biased voltage 1 V, R_on_ = 2 Ω, and L_on_ = 1.5 nH) in the off- and on-states. We perform full-wave numerical simulations with CST Microwave Studio 2019. The periodic boundary conditions of the unit cell are used along both the x- and y- axes, and the propagation of the incident wave is set along the z axis. The S-parameter retrieval method [33] can be used to extract the EM parameters. The conversion relationship between the equivalent impedance of the metadevice and the S-parameter is as follows:(1)$$Z\left(\omega \right)=\pm \sqrt{\frac{{\left(1+S_{11}\right)}^{2}-S_{21}{}^{2}}{{\left(1-S_{11}\right)}^{2}-S_{21}{}^{2}}}$$

When Re(Z) of the metadevice is below 0.1, there is a severe mismatch with the reference impedance, resulting in high reflectivity. Better impedance matching can be achieved when Re(Z) > 0.5 for transmission or absorption. When the equivalent impedance of the metadevice (Z_1_) and impedance in free space (Z_0_) match each other, the EM waves will pass through the interface without reflection and penetrate into the next layer. If the imaginary parts of equivalent permittivity and permeability are very small, the EM waves can pass through the material with low loss (wave-transparent material). Under the condition of impedance matching, further metallic shielding is introduced at the bottom (or metallic resonators to produce magnetic resonance between dual metallic layers) to obtain electrical or magnetic loss and, finally, perfect absorption can be achieved. At this point, the imaginary part is minimized and the real part of the impedance is close to unity. The total absorption of the metadevice can be simplified as:(2)$$A\left(\omega \right)=1-R\left(\omega \right)-T\left(\omega \right)=1-{\left|S_{11}\right|}^{2}-{\left|S_{21}\right|}^{2}$$

The “perfect” absorption depends on the transmittance T(ω) and reflectance R(ω) of the incident waves, and the absorption of x-polarized and y-polarized can be expressed as:(3)$$A_{x}=1-{\left|r_{xx}\right|}^{2}-{\left|r_{yx}\right|}^{2}-{\left|t_{xx}\right|}^{2}-{\left|t_{yx}\right|}^{2}$$
(4)$$A_{y}=1-{\left|r_{yy}\right|}^{2}-{\left|r_{xy}\right|}^{2}-{\left|t_{yy}\right|}^{2}-{\left|t_{xy}\right|}^{2}$$
where the subscripts x (y) denotes the x (y)-polarized waves. Accordingly, *r_xx_* and *r_yy_* (*t_xx_* and *t_yy_*) signify the co-polarized reflection (transmission) coefficients. Similarly, *r_yx_* and *r_xy_* (*t_yx_* and *t_xy_*) signify the cross-polarized reflection (transmission) coefficients.

**Figure 1 nanomaterials-12-04227-f001:**
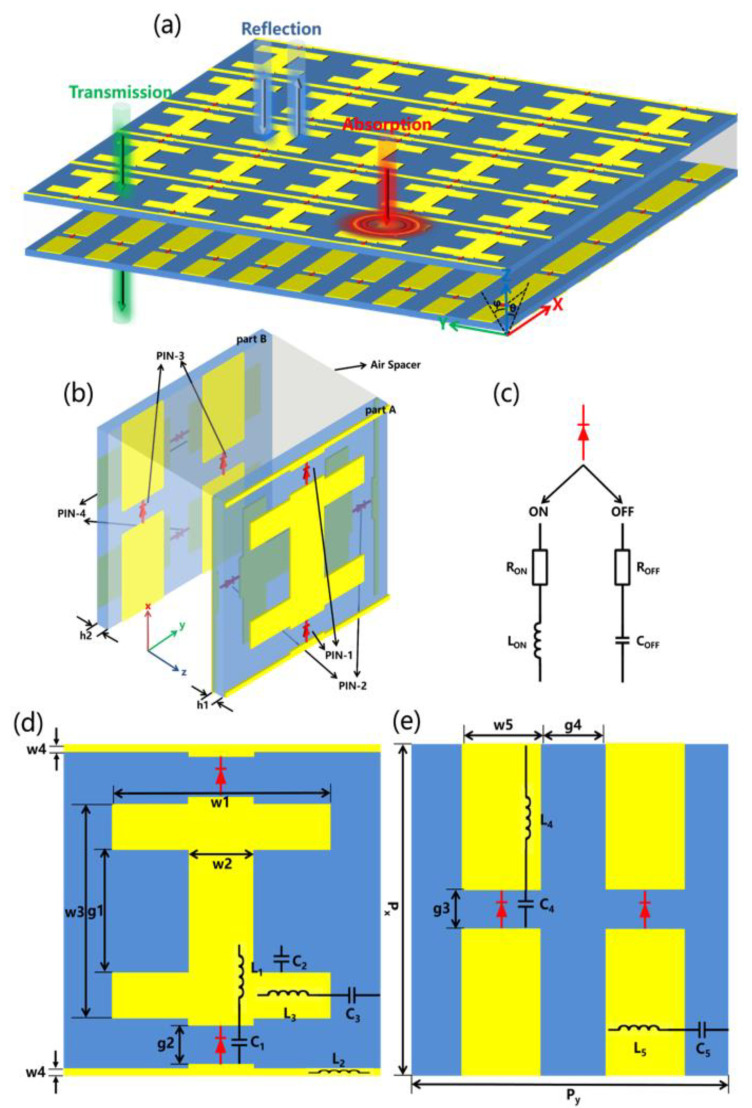
(**a**) The three-dimensional schematic of the proposed metadevice and illustration of EM functionalities; (**b**) Perspective view of the proposed meta-atom; (**c**) Equivalent circuit model of the PIN diode at the “on” and “off” states; (**d**,**e**) Top view of part A and part B.

## 3. Results and Discussion

Figure 2 depicts the transmission, reflection, and absorption spectra of the proposed metadevice. When PIN diode-1 and PIN diode-3 correspond to state-1 (on-state; on-state), the metadevice demonstrates a band-pass characteristic for x-polarized incident waves. The transmittance is higher than 80% in the range of 9.122–11.850 GHz. When PIN diode-3 is without bias, the metadevice exhibits a band-stop property in state-2 (on-state; off-state). At this time, the metadevice has a significantly reflective effect on incident waves and the reflectance is higher than 90% over a relatively wide band. With PIN diode-1 and PIN diode-3 operating in state-3 (off-state; off-state), the reflectance can be observed to drop to a minimum at 9.702 GHz and the transmittance is close to 0. At this point, the absorptivity reaches a peak of 97.9% (higher than 0.9 in the range from 9.205 to 10.741 GHz). Similarly, the metadevice can also perform the above three functionalities for y-polarized waves. It transmits signals (a transmittance higher than 0.8 from 9.130 to 11.366 GHz) when the PIN diode-2 and PIN diode-4 switch to state-1, whereas it blocks band signals (broadband high-efficiency reflection) when PIN diode-4 is shifted to the off condition. If the PIN diode-2 and PIN diode-4 are in state-3, the metadevice shows an almost perfect absorption effect at 10.098 GHz of approximately 97.6% (higher than 0.9 in the range from 9.658 to 10.895 GHz). Cross-polarization components of reflection and transmission are relatively small, which has no consideration here.

To further reveal the physical mechanism of the metadevice, we calculate the electric field and power loss density distribution at operating frequency (9.702 GHz). Only the x-polarized incident waves are analyzed here. The individual part A loaded on-state diodes are transparent for the EM waves. For state-1, the metadevice acts as an inductive equivalent circuit [25,29]. Both sides of the surface with on-state diodes do not resonate and the surface is transparent to the EM incident waves. The x-z plane electric field distribution at 9.702 GHz is shown in Figure 3a. Due to the driving of a strong electric field around part B, the electron oscillations generate strong induced currents, causing the electric field to radiate in the same direction and transmit EM wave energy as much as possible. In contrast, the metadevice behaves as a perfect electric conductor (PEC) when diodes correspond to state-2. As a result, the EM incident waves cannot pass through the metadevice. Figure 3b shows the electric field distribution in the x-z plane at 9.702 GHz. Almost no electric field components pass through part B, while the EM incident waves are reflected in the form of co-polarization. The above analysis shows that part B can be used as an EM switch by changing the external DC bias voltages of the diodes. According to impedance matching theory, EM waves will almost completely enter the absorber when the impedance is matched. Coupled EM wave energy will be eventually dissipated by the ohmic loss [34] and dielectric loss [34]. Here, the absorption is mainly caused by the ohmic loss, with almost no dielectric loss. Power loss distribution on the metadevice is mainly concentrated around PIN diode-1, and energy is dissipated in the form of joule heat, as shown in Figure 3c. Moreover, the absorption is not affected by the reduction in the loss tangents of dielectric (even if Rogers RO4003C is loss free), which also means that the absorption depends essentially on the ohmic loss, as shown in Figure 3d. The normalized complex impedance is derived from the complex reflection coefficient and complex transmission coefficient in Formula (1), as shown in Figure 3e. A range between 0.5 and 2 for the real part of relative impedance in transmission and absorption bands is marked in the diagram, indicating that an acceptable impedance match can be achieved in this area for either transmission or absorption. The impedance mismatch is severe when Re(Z) is less than 0.1, and a nearly total reflection can be achieved. The imaginary part of the relative impedance is close to zero at 9.702 GHz, and the real part is approximated to unity simultaneously, which eventually achieves perfect absorption. In other words, the equivalent impedance of the proposed absorber perfectly matches the free space, and the effect of absorption on this metadevice is superior.

These three functionalities are optimized and verified by the equivalent circuit method [30]. A simplified equivalent circuit of the metadevice under normal incidence is presented in Figure 4a. The lumped parameters corresponding to the specific structure are shown in Figure 1d,e. For part A, the top and bottom metallic layer are equivalent to a serial or parallel LC circuit, while each lumped diode is connected in parallel with a capacitor, where the C_i_ (i = 1, 2, 3) denotes the slot capacitance and L_i_ (i = 1, 2, 3) represents the effective inductance of the metallic wire gird and loop. Part B with metallic dipoles is equivalent to a parallel circuit consisting of two series LC circuits (L_4_C_4_ and L_5_C_5_) since each metallic dipole is equivalent to the series-connected inductor and capacitor [35]. The results for reflection coefficient S_11_ and transmission coefficient S_21_ can be obtained by converting the ABCD matrix [22] into S-parameters. Lumped element parameters are optimized by the advanced design system (ADS): C_1_ = 0.042 pF, C_2_ = 0.271 pF, C_3_ = 0.142 pF, C_4_ = 0.076 pF, C_5_ = 0.349 pF, L_1_ = 0.317 nH, L_2_ = 3.972 nH, L_3_ = 0.073 nH, L_4_ = 0.315 nH, and L_5_ = 0.091 nH. As depicted in Figure 4b, the calculated transmission, reflection, and absorption for the three states agree well with the simulated results, indicating that the equivalent circuit analysis is effective. The equivalent inductance and equivalent capacitance can be calculated by the following equations [30]:(5)$$L=\mu_{0}\left(p/2\pi \right)\log\left[\csc\left(\pi w/2p\right)\right]$$
(6)$$C=\varepsilon_{0}\varepsilon_{eff}\left(2l/\pi \right)\log\left[\csc\left(\pi g/2l\right)\right]$$
(7)$$\varepsilon_{eff}=\left(\varepsilon_{r}+1\right)/2$$

The dielectric substrate is modeled as a short section of transmission line with a length of h and a characteristic impedance of $Z_{sub}=Z_{0}/\sqrt{\varepsilon_{r}}$, where $\varepsilon_{r}$ and $\varepsilon_{eff}$ denote the relative permittivity and effective permittivity, respectively. The equivalent inductance is related to the structural period p and the wire width *w*, and the equivalent capacitance is related to the side length *l* and the metallic gap width *g*.

In addition, the sensitivity to the oblique incidence of these three functionalities is also one of the key factors that should be considered. Results for the oblique incidence of TE and TM mode plane waves are depicted in Figure 5. Transmission amplitude and bandwidth remain basically unchanged when the TE waves off-normal illuminate (up to 60°) the metadevice. Due to the high reflectivity of the TM waves, the transmission level tends to decrease sharply around 9.89 GHz. For the TE and TM modes, the reflectance is slightly degraded at large incident angles, as shown in Figure 5b,e. For the TE waves, the absorption amplitude and bandwidth remain almost the same as the incident angles increase. However, the absorption effect of the TM waves deteriorates (θ above 45°). This can be explained that the incident electric field in the TE mode can effectively induce strong resonant currents near two PIN diodes on the metallic plate to achieve ohmic loss. Although the resonant frequencies of absorption are slightly shifted, the amplitudes are still higher than 95%, which are given in Figure 5c,f. For the TE waves, the incident electric field is constant, while the equivalent magnetic field (the project component on the y-z plane) is decreased with the incident angles increased. The electric coupling effect between the incident electric field and the structure does not change. Therefore, the impact of the incident angles on the three functionalities is relatively small. On the contrary, the effective component of the incident electric field decreases in the case of the TM waves with oblique incidence, which leads to a decrease in the electric coupling effect and eventually deteriorates the performance. In short, the three functionalities possess a certain tolerance to wide-angle incidence. This also means that we can have a larger operating incident angle for future experiments.

The EM responses under the diversified combinations of lumped elements are studied in more detail below. Interestingly, there is a significant increase in absorption bandwidth when two PIN diode-1 are in opposite states, as shown in Figure 6a. The reflectance of −10 dB or less is taken as the evaluation criterion for absorption performance, and the absorptivity of the x-polarized wave is higher than 0.9 over the range of 9.449 to 11.847 GHz. Except for controlling the amplitude of the EM waves, its phase manipulation is also an important property. We replace the PIN diode-1 on the surface of the structure with a varactor (SMV2020-079LF). Its equivalent capacitance can be tuned at different reverse bias voltages to achieve phase manipulation of the reflected EM waves. Figure 6b shows the simulated reflection amplitude and phase curves of the designed metadevice under two configurations of varactor (state 0: V_R_ = 0 V and C_0_ = 3.2 pF; state 1: V_R_ = 20 V and C_1_ = 0.35 pF) [17]. The meta-atom can generate a phase difference of nearly 180° and a reflection amplitude of about 0.9 around 10.7 GHz. Two types of unit cells with 0 and π phase responses are expressed by the “0” and “1” elements, respectively. According to the coding concept [36], the phase distribution of each column is reasonably arranged, and our metadevice can generate novel EM functionalities (dual-beam scanning/multi-beam modulation), as given in Figure 6c. The splitting angle between two symmetrical radiation beams can be tuned by the periodic length of the coding sequences, and dynamic multi-beam radiation can also be generated by optimized random coding sequences. Further, many functionalities can be realized by manipulating the phase, such as anomalous refraction and anomalous reflection, according to generalized Snell’s law [8].

To validate the functionalities of the proposed metadevice, a prototype (size 41 × 41 mm^2^) with 2 × 1 meta-atoms was fabricated using an engraving machine (DCT DM350), as shown in Figure 7a,b. Considering the 90° rotational symmetry of the structure, we only measure the TE incidence. In the experiment, the waveguide (WR-90, 8.2–12.4 GHz) is connected to the network analyzer (ENA E5063A) through low-loss coaxial cables to measure the transmission level and reflection level of the sample. The DC power supplies (MOTECH LPS-305) are employed to provide the bias voltages for PIN diodes (the voltage of each on-state PIN diode is 1 V and each off-state PIN diode has no bias voltage). In addition, it is necessary to place insulation paper between the bias circuit and the inner wall of the waveguide to avoid short circuits in the ground. The measurement setup is presented in Figure 7c.

First, all diodes are in on-state and the transmission level of the metadevice is dominant, as shown in Figure 8a. The transmittance of the metadevice is higher than 0.7 at 9.239–11.582 GHz, and the reflectance always maintains a low level. When PIN-1 remains biased and PIN-3 is not biased, the reflection level of the metadevice maintains high in a wide frequency range, as shown in Figure 8b. Similarly, we set all diodes to the off-state without applying bias voltages. The reflectance can be observed to reach a minimum of 10.253 GHz and the transmittance is less than 0.1. At this point, the absorptivity reaches a peak of 97.8% (higher than 0.9 in the range from 9.907 to 10.561 GHz). Compared with the simulated results in Figure 2, it can be seen that the experimental results agree well with the simulated results. Due to the micromachining tolerance of the structure and the parasitic effect of diodes, a small frequency shift and amplitude fading can be observed. Other differences can be attributed to the incomplete alignment between the sample and the waveguide cavity.

## 4. Conclusions

To sum up, we present a reconfigurable metadevice that can dynamically switch between transmission, reflection, and absorption. Next, its EM responses are explained in terms of scanning parameters, electric field, and power loss. The transmission/reflection effect is switched by the on-/off-state of the diodes in part B, while the ohmic loss of the diodes in part A is the key factor leading to the absorption effect. The metadevice with diversified combinations of lumped elements is further studied in detail, and it is able to generate other EM functionalities (broadband absorption and dual/multi-beam modulation). This broadband, high-efficiency, and switchable microstructure endows this work with great potential in developing novel intelligent EM metadevices of communication fields.

## Figures and Tables

**Figure 2 nanomaterials-12-04227-f002:**
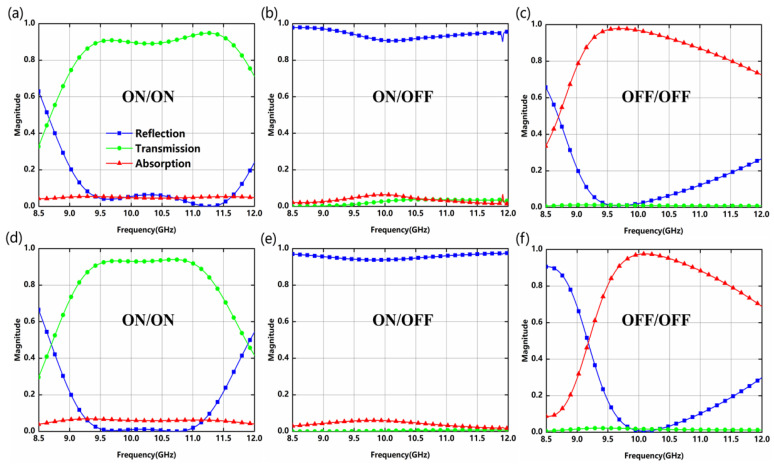
Transmission, reflection, and absorption for different states of the PIN diodes in the (**a**–**c**) x-direction and (**d**–**f**) y-direction.

**Figure 3 nanomaterials-12-04227-f003:**
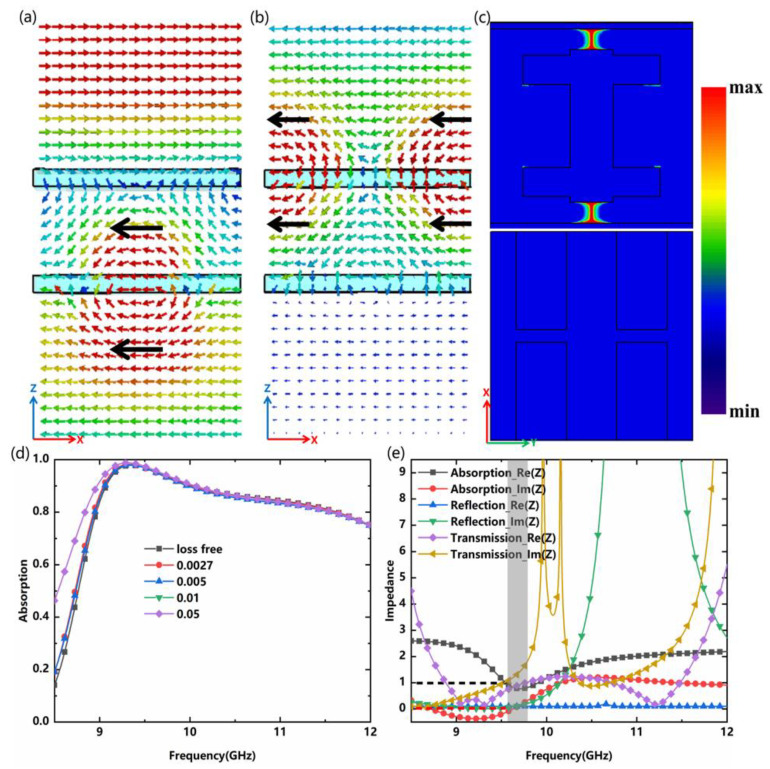
The x-z plane electric field distribution at 9.702 GHz of (**a**) Transmission mode; and (**b**) Reflection mode; (**c**) The power loss density distribution of part A and part B at 9.702 GHz; (**d**) The absorption corresponding to the dielectric with different loss tangents; (**e**) Real and imaginary parts of the complex impedance function.

**Figure 4 nanomaterials-12-04227-f004:**
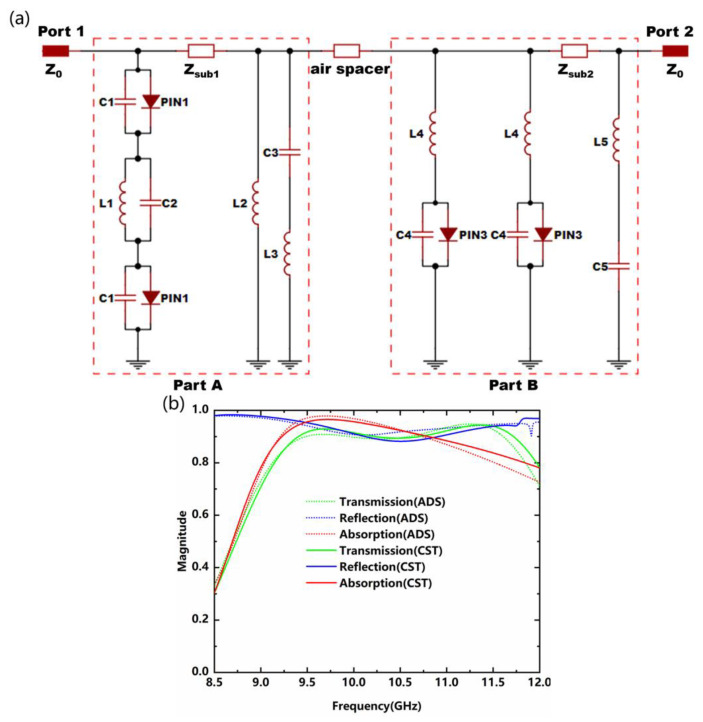
(**a**) Simplified equivalent circuit model of the proposed metadevice; (**b**) Calculation results of transmission, reflection, and absorption in three modes.

**Figure 5 nanomaterials-12-04227-f005:**
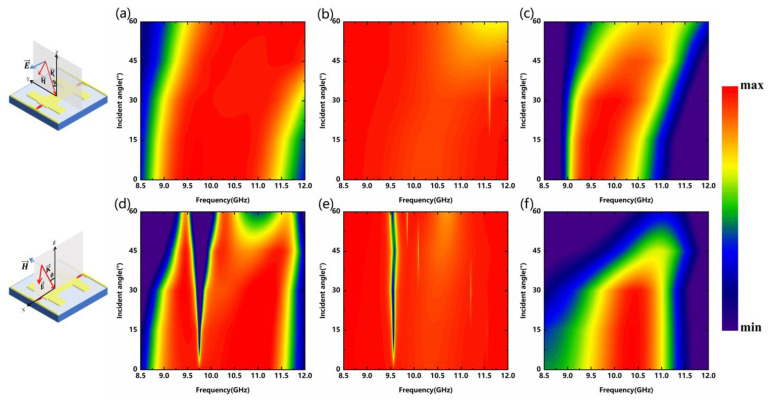
(**a**) Transmission spectra; (**b**) Reflection spectra; (**c**) Absorption spectra of the TE modes at different oblique incident angles; (**d**) Transmission spectra; (**e**) Reflection spectra; (**f**) Absorption spectra of the TM modes at different oblique incident angles.

**Figure 6 nanomaterials-12-04227-f006:**
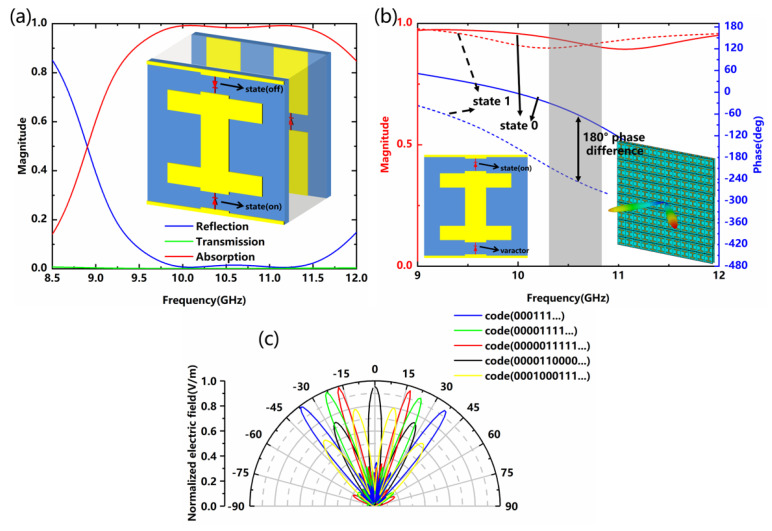
(**a**) Absorption effect under two PIN diode-1 in opposite states; (**b**) Simulated reflection phase and amplitude curves of the designed metadevice in state 0 and state 1 of the varactor; (**c**) Normalized electric field of the metadevice with different coding sequences.

**Figure 7 nanomaterials-12-04227-f007:**
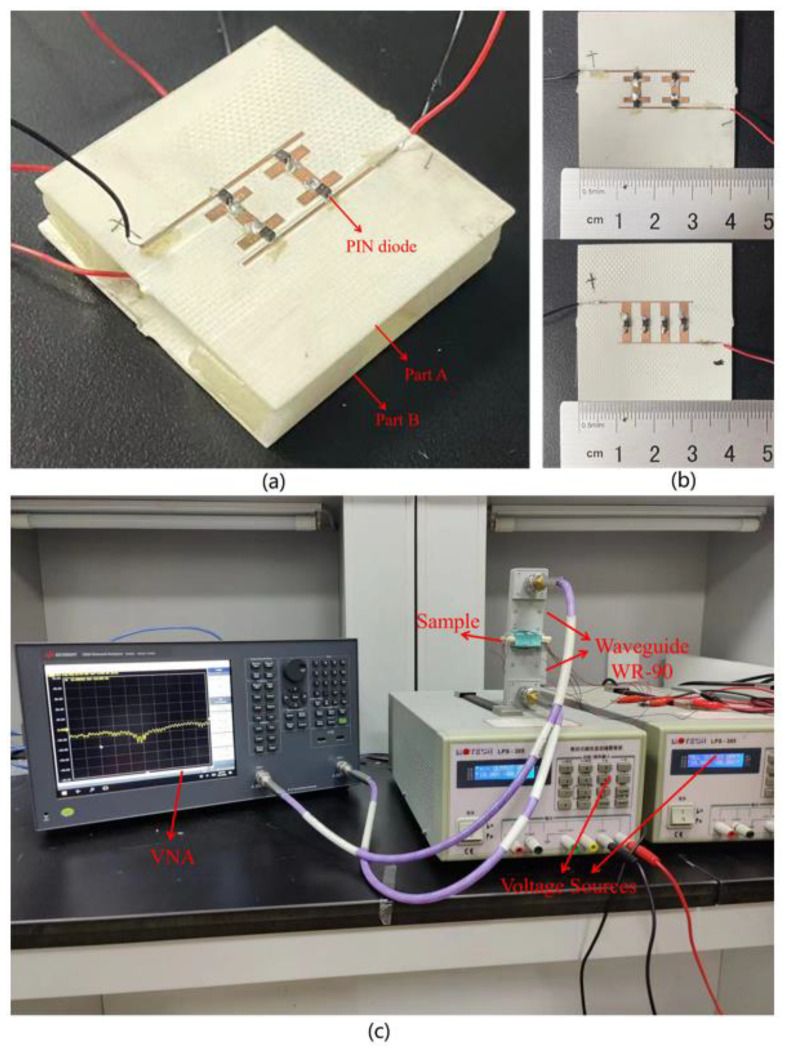
(**a**) Photographs of the fabricated prototype; (**b**) Top view of part A and part B; (**c**) Measurement setup.

**Figure 8 nanomaterials-12-04227-f008:**
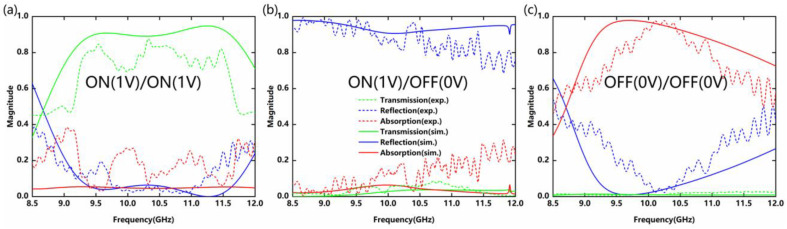
Simulated and measured reflection/transmission characteristics of the proposed metadevice under different operation modes (**a**) Transmission; (**b**) Reflection; and (**c**) Absorption.

## Data Availability

Experimental data from this study are available upon request.

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
