# Peer review of "An Amplitude-Modulated Metadevice with Switchable Reflection, Transmission, and Absorption"

_nanomaterials, 2022, doi:10.3390/nano12234227_

Round 1

Reviewer 1 Report

The manuscript presents a variation on the theme of diode-loaded metasurfaces. The problem posed (control of transmission/reflection/absorption and of polarization states) has been considered in the field for a while, and there have been numerous published papers (I would recommend citing papers by Tennant, such as DOI: 10.1109/LMWC.2003.820639). The proposed solution is a variation on the common theme of combining resonant elements with pin-diodes. While the structure itself is novel, the approach is not. The research methods are standard and sound. I would expect the paper if published to generate low impact. I am also questioning the choice of the journal, as there are no nanomaterials involved. 

Reviewer 2 Report

The paper is interesting and well written.

Reviewer 3 Report

The paper investigated numerically and experimentally switchable metasurface. The paper is basically interesting. However, there are some unclear points. Please address the following points.

1. As for Figs. 2, it seems to be polarization independent. However, the metasurface structure is asymmetric. Why does it have polarization independence?

2. The experiment was performed for 2 by 1 meta atoms. Why was 2 by 1 adopted? For example, Can one atom act as switchable metasurface?

3. As for Figs. 8, please add the relation between the numerical results such as Figs. 2.
